# Genetic Diversity of the Human Adenovirus C Isolated from Hospitalized Children in Russia (2019–2022)

**DOI:** 10.3390/v16030386

**Published:** 2024-02-29

**Authors:** Olga G. Kurskaya, Elena A. Prokopyeva, Nikita A. Dubovitskiy, Mariya V. Solomatina, Ivan A. Sobolev, Anastasiya A. Derko, Alina R. Nokhova, Angelika V. Anoshina, Natalya V. Leonova, Olga A. Simkina, Tatyana V. Komissarova, Alexey E. Tupikin, Marsel R. Kabilov, Alexander M. Shestopalov, Kirill A. Sharshov

**Affiliations:** 1Laboratory of Molecular Epidemiology and Biodiversity of Viruses, Federal Research Center of Fundamental and Translational Medicine, Novosibirsk 630060, Russia; kurskaya_og@mail.ru (O.G.K.); nikitadubovitskiy@gmail.com (N.A.D.); solomatina.mariyav@yandex.ru (M.V.S.); sobolev_i@hotmail.com (I.A.S.); a.derko19@gmail.com (A.A.D.); sharshov@yandex.ru (K.A.S.); 2Department of Children’s Diseases, Novosibirsk Children’s Municipal Clinical Hospital №6, Novosibirsk 630015, Russia; 3Department of Children’s Diseases, Novosibirsk Children’s Municipal Clinical Hospital №3, Novosibirsk 630040, Russia; olgasimkina83@yandex.ru (O.A.S.);; 4Genomics Core Facility, Institute of Chemical Biology and Fundamental Medicine, Novosibirsk 630090, Russia; alenare@niboch.nsc.ru (A.E.T.); kabilov@niboch.nsc.ru (M.R.K.)

**Keywords:** human adenovirus, adenovirus typing, genotypes C1, C2, C5, C89, C108, molecular epidemiology, pediatric infections, respiratory virus

## Abstract

The human adenovirus (HAdV) is a common pathogen in children that can cause acute respiratory virus infection (ARVI). However, the molecular epidemiological and clinical information relating to HAdV among hospitalized children with ARVI is rarely reported in Russia. A 4-year longitudinal (2019–2022) study among hospitalized children (0–17 years old) with ARVI in Novosibirsk, Russia, was conducted to evaluate the epidemiological and molecular characteristics of HAdV. Statistically significant differences in the detection rates of epidemiological and virological data of all positive viral detections of HAdV were analyzed using a two-tailed Chi-square test. The incidence of HAdV and other respiratory viruses such as human influenza A and B viruses, respiratory syncytial virus, coronavirus, parainfluenza virus, metapneumovirus, rhinovirus, bocavirus, and SARS-CoV-2 was investigated among 3190 hospitalized children using real-time polymerase chain reaction. At least one of these respiratory viruses was detected in 74.4% of hospitalized cases, among which HAdV accounted for 4%. A total of 1.3% co-infections with HAdV were also registered. We obtained full-genome sequences of 12 HAdVs, which were isolated in cell cultures. Genetic analysis revealed the circulation of adenovirus of genotypes C1, C2, C5, C89, and 108 among hospitalized children in the period from 2019–2022.

## 1. Introduction 

The human adenovirus (HAdV) has double-stranded DNA and a non-enveloped icosahedral structure, which includes three viral capsid proteins: hexon, penton base, and fiber [1]. HAdV belongs to the genus Mastadenovirus (family Adenoviridae) [2]. Currently, seven different species of HAdV (from A to G) have been identified [2], while, according to the classification criteria of the Human Adenovirus Working Group, 115 HAdV genotypes have been assigned [3].

Adenovirus infection in children may cause respiratory illnesses (bronchitis, laryngotracheitis, tracheitis, rhinopharyngitis, pharyngitis, and pneumonia) or other disease complications (otitis, conjunctivitis, gastrointestinal symptoms, hemorrhagic cystitis, mesenteric adenitis, myocarditis, and encephalitis) [4,5,6,7]. It is believed that among all cases of acute respiratory viral infection (ARVI) in children, HAdV accounts for at least 5–10% [8], and such species as HAdV-C (namely, genotypes C1, 2, 5, 6, 57, 89, 104, 108) may be associated with local ARVI outbreaks among children [9,10]. In Russia, the incidence of HAdV was 6.9% in children aged 0–17 years during the period 2004–2014, with co-infections occurring in 41% of cases [11]. Among all types of adenovirus, the proportion of HAdV-C in children under 1-year-old was 60%, in preschoolers (1–7 years old)—53.6%, and in school children (7–17 years old)—21.4% [11]. According to Yatsishina et al. [11], in Russia during the period 2004–2014, the most common HAdV etiological agents among children were HAdV-C2 (20.9%) and HAdV-C6 (11.6%), the average age with adenovirus infection in children being 3 years and 11 months. 

In Russia, knowledge of the epidemiology and genotyping of HAdV-C infection is currently lacking. Our results present the virological, epidemiological, and genotyping data of HAdV-C arising from a retrospective observational study that collected laboratory-confirmed HAdV cases in the period 2019–2022 at the two Novosibirsk Children’s Municipal Clinical Hospitals №6 and №3.

Here, we report the molecular epidemiological and virological characteristics of HAdV with C1, C2, C5, C89, and C108 detected in Russia among 3109 samples from children 0–17 years old. Overall, the study sheds light on the epidemiological situation and viral characteristics of HAdV-C and its circulating genotypes in one region of Russia. 

## 2. Materials and Methods 

### 2.1. Ethics Issues

All aspects of the study were approved by the Committee on Biomedical Ethics of the Federal Research Center of Fundamental and Translational Medicine Protocol № 3 on 28 January 2019. Written informed consent was obtained from all parents/legal guardians prior to sample taking.

### 2.2. Sample Collection

Nasal and throat swabs were taken from children (aged 0–17 years) hospitalized with symptoms of acute respiratory infections during three successive research periods: the first being 2019–2020 (November–April), which was the pre-pandemic season; the second was 2020–2021 (November–April); and the third was 2021–2022 (October–April). Both the latter were seasons of the continuing COVID-19 pandemic. We enrolled children who had at least one of the systemic symptoms (fever, headache, myalgia, or malaise) and one of the respiratory symptoms (cough, rhinorrhea, nasal congestion, sore throat, shortness of breath, lung auscultation abnormalities, or chest pain). Exclusion criteria for samples were the disease duration for more than 7 days. Samples were taken in two hospitals, Novosibirsk Children’s Municipal Clinical Hospital №6 and Novosibirsk Children’s Municipal Clinical Hospital №3. Samples were placed in tubes with transport medium (Dulbecco’s modified Eagle’s medium (Capricorn Scientific, Germany) with 0.5% bovine serum albumin, 100 μg/mL of gentamicin sulfate (BioloT, Saint Petersburg, Russia), and 50 units/mL of amphotericin B (BioloT, Saint Petersburg, Russia) and were stored at 2 °C to 8 °C before analysis but not more than 48 h.

### 2.3. Virus Detection

Nasal and throat swabs were tested using real-time polymerase chain reaction (RT-PCR). Viral nucleic acids were extracted from all clinical samples using an RNA/DNA extraction kit «RIBO-sorb» (Interlabservice, Russia) according to the manufacturer’s instructions. The reverse transcription of extracted viral nucleic acids was immediately performed using the commercial kit “REVERTA-L” (Interlabservice, Russia). The following RT-PCR kits were used: AmpliSens ARVI-screen-FL and AmpliPrime Influenza SARS-CoV-2/Flu(A/B/H1pdm09, Interlabservice, Russia) for adenovirus (HAdV), SARS-CoV-2 and influenza A and B viruses (HIFV), and other respiratory viruses (respiratory syncytial virus (HRSV); alphacoronaviruses (NL63/229E) and betacoronaviruses (OC43/HKU1) (HCoV); parainfluenza virus types 1–4 (HPIV); metapneumovirus (HMPV); rhinovirus (HRV); and bocavirus (HBoV)). Positive and negative controls were included in each run. Each kit of reagents was targeted to a conserved site of the viral genome and was not specified in the instructions as the intellectual property of the company. The visualization of co-infections was created with the UpSetR package v.1.4.0 [12].

### 2.4. Cells

HEp-2, Vero, HeLa, Caco-2, and SPEV (cell culture collection of State Research Center of Virology and Biotechnology «VECTOR», Russia) were maintained in Dulbecco’s modified Eagle’s medium (Capricorn Scientific, Germany), 10% fetal bovine serum (FBS) (Capricorn Scientific, Germany), and 50 μg/mL of gentamicin sulfate (BioloT, Russia). All cells were incubated at 37 °C, 5% CO_2_.

### 2.5. Virus Isolation 

After receiving the positive result for HAdV in real-time PCR, virus isolation on HeLa, HEp-2, Vero, SPEV, and Caco-2 cells was carried out. In total, 96-well plates with a daily monolayer of HeLa, Hep-2, Vero, SPEV, and Caco-2 cells in a growth medium were washed with a Henks solution and then inoculated with 30 μL aliquots of the clinical sample. After 1 h of incubation at 37 °C, 5% CO_2_ supernatant was removed, and 200 μL/well of DMEM (Capricorn Scientific, Germany) with 2% of FBS (Capricorn Scientific, Germany) and 100 μg/mL of gentamicin sulfate (BioloT, Russia) were added. Intact HeLa, HEp-2, Vero, SPEV, and Caco-2 cells were used as controls in each plate. Infected plates were incubated at 37 °C, 5% CO_2_ for up to 7 days, and were checked daily under the microscope (Micromed I, Russia) by the presence of a cytopathogenic effect (CPE) manifested by cell death.

### 2.6. Light and Electronic Microscopy and Immunocytochemical Analysis

Inoculated cells were fixed with a 4% formalin solution on days 4–5 post infection. After that, the cells were stained with azur and eosin for 30 s (Minimed, Russia). Evaluation of CPE was estimated under an inverted AxioVert 40 microscope (Carl Zeiss, Germany) with AxioCam ICc 3 camera (Carl Zeiss, Germany), Bright Field mode. The CPE score was assessed by the presence of cytopathic changes in the cell monolayer, the destruction of the cell monolayer, the formation of clusters of rounded apoptotic cells, the loss of intercellular contacts, and the rounding of the cell shape. Intact HEp-2 and Vero cells were used as controls to compare the morphological state of inoculated and intact cells monolayer.

Localization of HAdV in the infected cells was revealed by immunofluorescence (IF) assay [13]: cells on the coverslips were fixed with a 4% formalin solution for 10 min and were treated by diagnostic kit with FITC-labeled fluorescent antibodies to HAdV (LLC “Enterprise for the production of diagnostic drugs”, St. Petersburg, Russia). The presence of HadV was identified by a green immunofluorescent signal in the cells. The cell nuclei were contrasted with 5µM 4,6-diamino-2phenyl indole (DAPI), briefly washed with deionized water, and mounted in the Prolong Antifade/Slow Fade mounting medium. Cells were imaged using an LSM710/NLO confocal microscope (Carl Zeiss, Germany), and images were processed with Adobe Photoshop CS6 software. The presence of HAdV was identified by a green immunofluorescent signal in the cells. 

For electronic microscopy, a 200-mesh copper TEM grid (SPI Supplies, West Chester, PA, USA) with ultrathin (invisible on the water surface) [14] formvar support film was placed on a 10 μL droplet of Hela cell suspension infected with HAdV for 30 s and then dried using filter paper. A 10 μL droplet of uranyl acetate water solution (1% w/v) was then added to the grid surface for 15 s. After drying, using filter paper again, the grid was examined with a TEM (JEM-1400; Jeol, Japan) at an accelerating voltage of 80 kV. 

### 2.7. Sequencing

DNA was isolated by using a GeneJET viral DNA/RNA purification kit (Thermo Fisher Scientific, Waltham, MA, USA) and fragmented to an average size of 400 bp in a microTUBE AFA fiber snap-cap tube (Covaris S2 instrument). The paired-end (PE) library was constructed using dual-index NEBNext multiplex oligonucleotides and NEBNext Ultra II DNA library prep kit for Illumina (New England BioLabs). The DNA library was sequenced with a reagent kit v3 (600 cycles) on a MiSeq sequencer (Illumina) in the Genomics Core Facility (ICBFM SB RAS). The reads were quality trimmed, and adapter sequences were removed using Cutadapt v3.7 [15]. Full-length genomes were assembled de novo with SPAdes v3.15.3 [16]. The genome sequences of HAdV were deposited in GISAID. Accession ID: MZ151860 - HAdVC/Novosibirsk/8.65Hp/2020; MZ151861 - HAdVC/Novosibirsk/7.134V/2019, MZ151862.1 - HAdVC/Novosibirsk/8.234V/2020; MZ151863.1 - HAdVC/Novosibirsk/8.202V/2020; MZ151864.1 - HAdVC/Novosibirsk/8.142V/2020; MZ151865.1 - HAdVC/Novosibirsk/8.171V/2020; MZ151866 HAdVC/Novosibirsk/8.81V/2020; ON152649 - HAdVC/Novosibirsk/7.2Hp/2022; ON152650 - HAdVC/Novosibirsk/7.273Hp/2022; ON152651 - HAdVC/Novosibirsk/7.17Hl/2021; ON152652 - HAdVC/Novosibirsk/7.45Hl/2021; ON152653 - HAdVC/Novosibirsk/8.135Hl/2021.

### 2.8. Sequence Analysis

Human adenovirus sequences being investigated were combined with sequences retrieved from the GenBank database. For multiple alignments, a MUSCLE software was used [17]. Comparative pairwise sequence alignment of 12 investigated viruses was performed via BioEdit [18]. Phylogenetic trees were built via MEGA 11 using maximum likelihood following results of the best-fit model test in MEGA (complete genome sequence – GTR + G + I; penton – T92 + G + I; hexon – TN93 + G; fiber – HKY + I) [19]. Bootstrap support values were generated using 1000 bootstrap replicates. SimPlot analysis was performed with SimPlot++ v.1.3 [20]. Features of the 8.171V strain genome were visualized with DNA Feature Viewer.

### 2.9. Statistical Analysis 

Data analysis was performed using Microsoft Excel 16.16.2 for MacOS and GraphPad 9.1.1 (GraphPad Software). Descriptive statistics included mean and standard error (for quantitative traits) and absolute and relative proportion (for qualitative traits). Categorical data were analyzed using a two-tailed Chi-square test undertaken to compare infection rates for respiratory viruses among different age groups. *p*-value < 0.05 was considered to be statistically significant.

## 3. Results

### 3.1. Epidemiology

During the period 2019–2022, a total of 3190 samples were collected from children aged 0–17 years who were hospitalized with ARI at the Novosibirsk Children’s Municipal Clinical Hospital №6 and the Novosibirsk Children’s Municipal Clinical Hospital №3. Among these children, 1742 (54.6%) were males and 1448 (45.4%) were females (Table 1). No significant gender differences in morbidity among these patients were observed. At least one of the respiratory viruses (HAdV, HIFV, HRSV, HCoV, HPIV, HMPV, HRV, HBoV, and SARS-CoV-2) was revealed in 74.4% (2372/3190) of samples by PCR analysis, whereas the viral co-infection was detected in 8.7% (277/3190) (Table 1). The most numerous cohort of infected children was 0–2 of age, amounting to 68% (1248/1836) (Chi-square test, *p* < 0.01), while the school-aged cohort showed the least percent of viral infections, which was 53% (267/502) (Chi-square test, *p* < 0.01) (Table 1). The level of viral co-infection also decreased among children older than 2 years, reaching a minimum proportion of 2% (9/502) in the age group of >7 years. Among the youngest children (<2 years), we detected co-infections in 10% (175/1836) of cases.

Among the 3190 hospitalized children, 3.98% (127/3190) tested positive for HAdV (Table 1). The lowest level of HAdV-positive cases (1.6%, 8/502) was in children aged 7–17 years. In 1.3% of the cases overall, HAdV was found in combination with other respiratory viruses, namely HIFV, HRSV, HCoV, HPIV, HMPV, HRV, or SARS-CoV-2 (Table 1, Figure 1). It is interesting to note that HAdV was detected in triple infections as follows: HAdV + HCoV + HPIV, HAdV + HRSV + HCoV, and HAdV + HRSV + SARS-CoV-2.

### 3.2. Virus Isolation and Virological Characteristics

We randomly selected 30 samples out of 127 PCR positives with the lowest Ct values (18–23), representing different ages and sexes, which we attempted to isolate in cell cultures. In order to study virological characteristics, five cell cultures of HeLa, HEp-2, Vero, SPEV, and Caco-2 were infected with the first human adenovirus (strain HAdVC/Novosibirsk/7.134V/2019) isolated at the beginning of the research period—in December 2019. Cells were evaluated daily, and on the 4th or 5th day of the appearance of CPE, they were fixed and stained for further analysis under light or confocal microscopes. The results showed the same morphological changes in studied cell cultures, characterized by an apoptotic type of destruction on the 4th or 5th day post infection (Figure 2 and Figure 3).

Because the observed cytopathic effect of human adenovirus was similar in all five studied cell cultures, we decided to use cell cultures most commonly employed as laboratory tools: Vero, HEp-2, or HeLa cell cultures—for the further isolation of adenoviruses.

The identification of human adenovirus by isolation on cell cultures was confirmed by PCR and immunofluorescence. IF analysis of cells infected with HAdVs showed specific fluorescence of the cells (Figure 4). The morphology of virions was confirmed by electron microscopy. The morphology typical of adenovirus—icosahedral non-enveloped capsid (diameter 70–90 nm)—was observed (Figure 5).

Thus, 12 isolates showed active replication with extensive CPE and were processed for NGS with de novo assembly. 

### 3.3. Phylogenetic and Genomic Analyses

The complete genomes of twelve HAdV isolates were successfully sequenced and deposited in the NCBI Genbank (accession numbers are indicated in the Section 2). Then, we conducted the phylogenetic analysis of these HAdV isolates and other representative HAdVs available from the Genbank database. Phylogenetic analysis based on the complete genome revealed that our isolates belonged to five genotypes: C1, C2, C5, C89, and C108 (Figure 6; Table 2 and Table 3; Appendix A). According to the phylogenetic analysis of the complete genome, as well as additional analyses undertaken separately made for the fiber gene, hexon gene, and penton base gene, 3 strains belonged to the C1 genotype (HAdVC/Novosibirsk/8.65Hp/2020, HAdVC/Novosibirsk/8.81V/2020, and HAdVC/Novosibirsk/8.135Hl/2021) and 5 strains to genotype C2 (HAdVC/Novosibirsk/8.142V/2020, HAdVC/Novosibirsk/7.134V/2019, HAdVC/Novosibirsk/8.202V/2020, HAdVC/Novosibirsk/7.17Hl/2021, and AdVC/Novosibirsk/7.45Hl/2021). The sequences of two strains of the C1 genotype isolated in 2021 and 2022 (HAdVC/Novosibirsk/8.81V/2020 and HAdVC/Novosibirsk/8.135Hl/2021, respectively) were closely related phylogenetically to each other. Two strains isolated in 2022 belonged to the C5 genotype (HAdVC/Novosibirsk/7.2Hp/2022 and HAdVC/Novosibirsk/7.273Hp/2022) (Figure 6; Table 2 and Table 3; Appendix A).

In addition, we found two strains of the twelve isolates studied to be of interest. Strain HAdVC/Novosibirsk/8.234V/2020 was determined to belong to the C89 genotype, according to the phylogenetic tree referenced and based on the complete genome, and was closely related to a strain isolated from China (Accession number MH121097.1) (Figure 6). This strain had fiber and hexon genes belonging to the C2 genotype and pentone base genes of the C89 clade.

Another strain, HAdVC/Novosibirsk/8.171V/2020, had the highest nucleotide identity with C108 genotype HAdV sequences (OQ108498, 99.61%; ON054624, 99.89%) (Table 3, N4). Its complete genome sequence formed a large clade with other C108 genotype strains isolated mainly in Asia. The penton base of this genotype strain belongs to the C1 genotype, while hexon and fiber genes relate to the C2 genotype. SimPlot analysis showed that its nucleotide sequence has higher identities for the C108 sequence than C2; hence, the 1–8000 region has a lower identity score with the C108_1 reference sequence than with the most related C108_2 sequence (Figure 7A,B). 

In general, the strains of human adenovirus C studied formed several groups according to the pairwise distances of full-genomic nucleotide sequences, as confirmed by the similarity of genotypes 1, 2, 5, 89, and 108 (Table 2; Appendix A).

All strains studied had similar virus titers ranging from 3.25 to 3.75 lgTCID_50_/_mL_ (Table 3). Thus, we did not observe differences in virus growth between the different genotypes.

## 4. Discussion

In our study, we analyzed certain molecular, epidemiological, and virological characteristics of HAdV with C1, C2, C5, C89, and C108 detected from children 0–17 years old. The human adenovirus is a pathogen commonly associated with acute respiratory infections among hospitalized children. In conformity with the officially determined epidemic season in the Russian Federation, all samples for this study were collected from October–November to March–April during the period 2019–2022. The incidence of ARVs (HAdV, HIFV, HRSV, HCoV, HPIV, HMPV, HRV, HBoV, and SARS-CoV-2) was analyzed by real-time PCR among 3190 hospitalized patients 0–17 years old with acute respiratory infections in the city of Novosibirsk, Russia. At least one of these respiratory viruses was detected in 74.4% (2372/3190) of the hospitalized children. The lowest incidence of ARIs was registered among patients older than seven (Table 1), as also determined in earlier works [21,22,23]. Those children 0–2 years old were the most sensitive to viral infections, with a 68% positive rate (1248/1836) (Chi-square test, *p* < 0.01). The most common etiological agent of ARI among children during the 2019–2020 period was HIFV 28.7% (312/1088); during the 2020–2021 period was HMPV 28% (316/1130); and during the 2021–2022 period was HRSV 20.3% (197/972). The most common etiological agent of ARI among children during the period 2019–2020 was HIFV 28.7% (312/1088); during the period 2020–2021 was HMPV 28% (316/1130); and during the period 2021–2022 was HRSV 20.3% (197/972). In our previous study, the absence of HIFV among circulated viruses during the period 2020–2021 was shown, as well as the absence of HMPV among other respiratory viruses during the subsequent period (2021–2022) [21]. In total, as shown in our previous work, the assessment reports about etiological agents of ARVI during the period 2019–2022 that HAdV presented at a lower rate, following HIFV, SARS-CoV-2, HRSV, and HMPV [21,22].

In the study reported here, 3.98% of pediatric patients with ARI proved to be HAdV positive. Approximately similar data were obtained in China (3.78–7.0%) during the period 2017–2019 [9,22,24]. In our current research, the majority of the HAdV-positive cases (4%, 82/1836) were detected among children younger than 2 years. A similar epidemic pattern was observed in Europe and Asia [22,25,26,27]. In total, co-infections of human AdVs with any other respiratory viruses were registered in 1.3% of cases during the period 2019–2022. In our previous work [21], it was shown that HAdV ranked in fifth place among etiological agents in the frequency of co-infection cases, such as HPIV, HCoV, HRV, HMPV, and HBoV during the period 2019–2022. HAdV was detected in triple infections as follows: HAdV + HCoV + HPIV, HAdV + HRSV + HCoV, and HAdV + HRSV + SARS-CoV-2.

Initially, we set the goal to identify isolates using various cell cultures. We randomly selected 30 samples with the lowest Ct values (18–23), which represented different age groups and sexes, and attempted to isolate them. As a result, 12 isolates showed active replication in cell cultures, and as a consequence, it was decided to conduct NGS sequencing and undertake a genome analysis since such data did not exist for this region of Russia.

It is interesting to note that human adenovirus C may be isolated at least on five cell cultures—HeLa, HEp-2, Vero, SPEV, and Caco-2—studied in this research. The average viral titers shown on different cell cultures were in diapason 3.25–3.75 lgTCID_50_/mL. Virological characteristics did not differ between the five genotypes of HAdVs detected. In comparison with the results of Biliavska et al. [28], the viral activity of our isolated HAdVs isolated on cell cultures was lower. Through electron microscopy, we confirmed the standard icosahedral form characteristic of adenoviruses.

The complete genome sequences and phylogenetic analysis of the twelve HAdV isolates revealed that all of them belonged to the C species. In this study, the maximum likelihood trees and pairwise differences in human adenovirus sequences were used to evaluate genotypic relationships and paired with the epidemiological data from routine infection prevention and control records and hospital activity data. Here, we have shown the results of the sequences and phylogenetic analysis of the complete genome, fiber, hexon, and penton base genes of the HAdVs isolated. The research indicates that various evolutionary variants of HAdVs were circulating in Novosibirsk during the period of this research. We detected several genotypes of HAdV—C1, C2, C5, C89, and C108—that were circulating during the period 2019–2022. However, such data do not allow us to make conclusions concerning the genotype prevalence, as we have not yet determined the complete genotype for all PCR-positive samples. Such work in the future will allow us to determine in more detail the epidemiology of adenoviruses and to assess whether there is a difference in clinical manifestations and disease severity. This will be of great importance for practical medicine.

In the countries closest geographically to Russia, the most prevalent species of HAdV reported were as follows: genotypes C1, C2, and C5 in Japan during the period 2008–2015 [25] and HAdV-C1,2,5 during the periods 2011–2018 and 2021–2022 [29,30], and HAdV-C2 in China during the period 2017–2018 [31,32]. The HAdV-C89 genotype was also recorded sporadically in China [10,33].

Despite the fact that we cannot support the conclusions about the genotypes prevalent in our study because of the limited number of genotyped samples, we showed evidence of the circulation of various HAdV variants. For the first time, the circulation of the HAdV-C89 and C108 genotypes in Russia was shown, according to the results of phylogenetic analysis based on penton base, fiber, hexon nucleotide sequences, and complete genomes. The phylogenetic clade of HAdVC/Novosibirsk/8.234V/2020 includes strains 29C2 and 47C2, according to a recent study in Germany [10], where these strains were assigned to the new C-89 genotype, based on differences in penton base gene sequences. Phylogenetic analyses of other isolates through fiber and hexon gene sequencing showed seven strains clustered with HAdV-C2, three strains clustered with HAdV-C1, and two with HAdV-C5. It is noteworthy that the presence of the HAdV-C108 genotype strain (HAdVC/Novosibirsk/8.171V/2020) has also been described for the first time in Russia in our study. According to phylogenetic analysis, it has the typical genetic features of this particular genotype—P1H2F2.

### Limitations

It should be noted that there are some limitations in our study. Our study sample was obtained only from hospitalized patients. It is also known that the state of virus infection of a carrier can be asymptomatic. Accordingly, the distribution of viruses among the Russian population may differ from our sample results. We also did not analyze social and behavioral factors that may serve as potential drivers of viral interference, and we cannot exclude the possibility of surveillance artifacts. The strength of our data is based on sequential observations from both pre-pandemic and pandemic periods through research spanning three years and the collection of samples from 3190 patients 0–17 years old. Annual correct monitoring of seasonal fluctuations of common respiratory viruses has the potential to improve general knowledge about viral interactions. However, as global surveillance systems for non-influenza respiratory viruses and for non-COVID-19 research remain limited in both scope and funding, the ability to undertake such investigations remains challenging.

## 5. Conclusions

The study of the incidence of HAdV and other respiratory viruses, namely HIFV, HRSV, HCoV, HPIV, HMPV, HRV, HBoV, and SARS-CoV-2 among 3190 hospitalized children showed that at least one of these respiratory viruses was detected in 74.4% of hospitalized cases, among which HAdV accounted for 4%, whereas co-infections with HAdV among 1.3 %. We obtained full-genome sequences of twelve HAdVs isolated in cell cultures. Genetic analysis has demonstrated that at least five genotypes (C1, C2, C5, C89, and C108) of the HAdV virus were circulating among children (0–17 years old) with ARVI in Novosibirsk, Russia, during the period 2019–2022. We have revealed the circulation of 89 and 108 genotypes for the first time in Russia. Further comprehensive and systematic monitoring, detection, and characterization of HAdV are necessary for future research and clinical practice. A future integrated approach that includes clinical data and obligatory full-genome analysis will provide the opportunity to explore the clinical significance of different genotypes, understand the implications of co-infections, and further investigate yet unassigned strains. Data on differences in clinical manifestation and disease severity for certain genotypes will be of great importance for practical medicine.

## Figures and Tables

**Figure 1 viruses-16-00386-f001:**
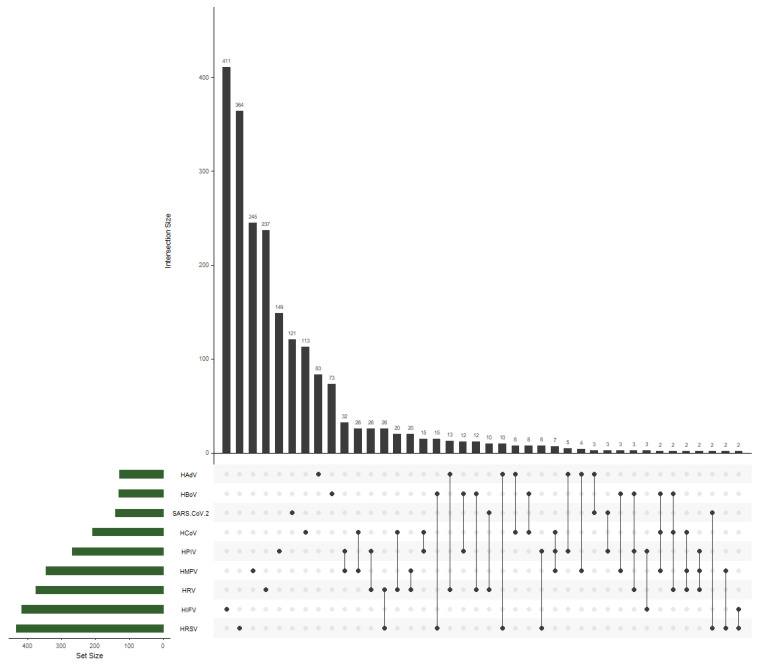
Acute respiratory virus infections and their combinations detected in samples from patients aged 0–17 years hospitalized during the period 2019–2022. Note: Dots—combinations or individual infections with dark grey bars visualizing number of cases in different combinations. Only cases ≥ 2 included.

**Figure 2 viruses-16-00386-f002:**
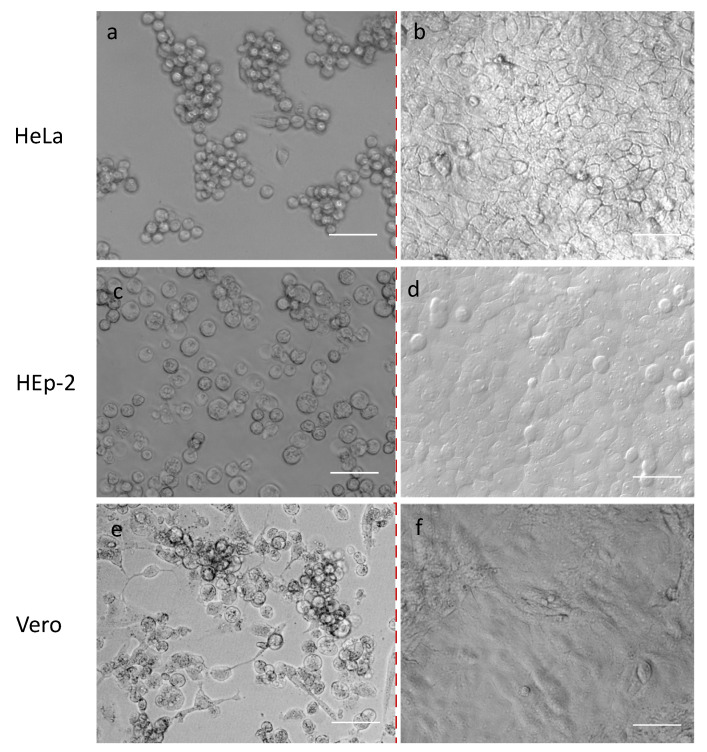
Cytopathogenic effect of HAdV on HeLa, HEp-2, and Vero cell cultures. Note: (**a**,**c**,**e**)—infected with HAdV HeLa, HEp-2, and Vero cell cultures, respectively; (**b**,**d**,**f**)—control cells of HeLa, HEp-2, and Vero cell cultures, respectively. Scale Bar 75 µm.

**Figure 3 viruses-16-00386-f003:**
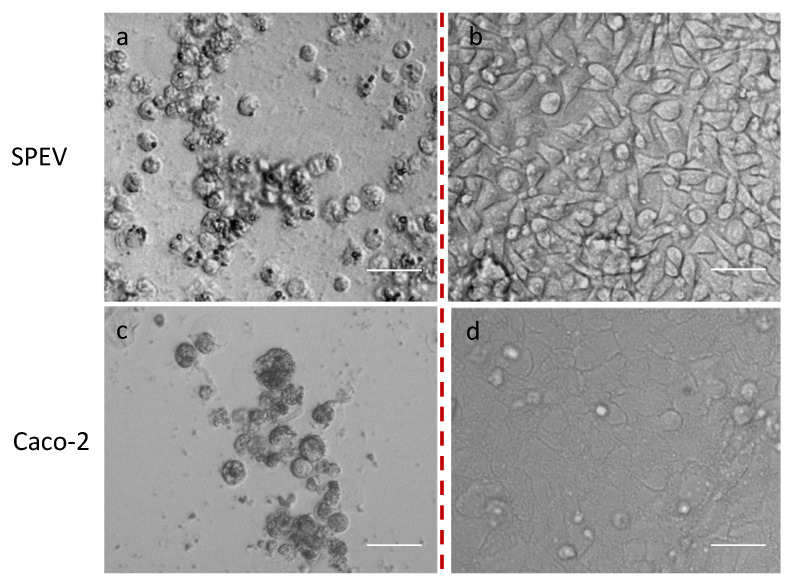
Cytopathogenic effect of human adenovirus on SPEV and Caco-2 cell cultures. Note: (**a**,**c**)—infected with human adenovirus SPEV and Caco-2 cell cultures, respectively; (**b**,**d**)—control cells SPEV and Caco-2 cell cultures, respectively. Scale Bar 75 µm.

**Figure 4 viruses-16-00386-f004:**
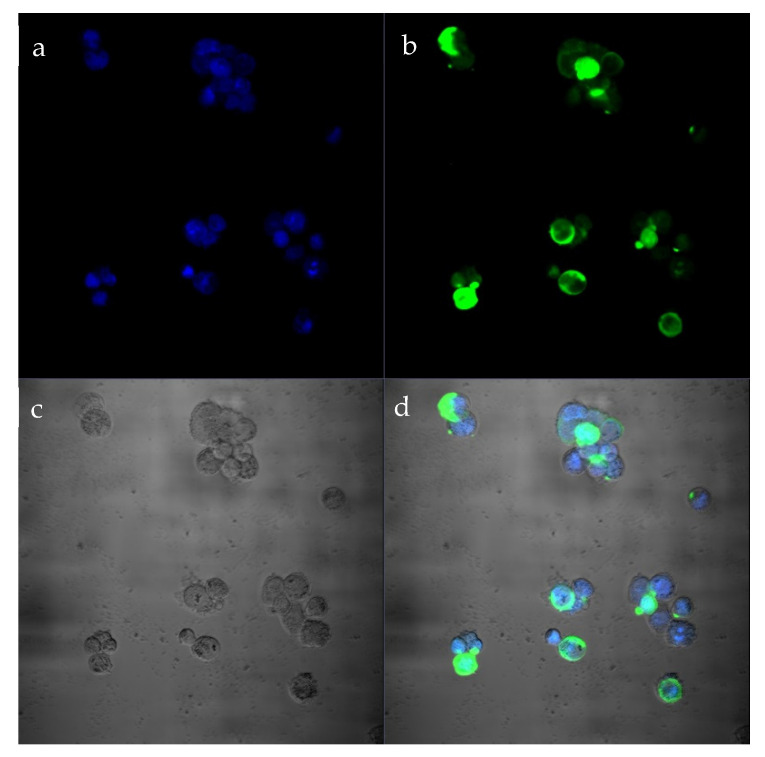
Immunofluorescent analysis of HeLa cell cultures infected with human adenovirus. Note: HeLa cells were infected with the human mastadenovirus C (strain HAdVC/Novosibirsk/7.134V/2019), incubated for 5 days, then fixed and stained using a diagnostic kit with fluorescent FITC-labeled antibodies to HAdV (Enterprise for the Production of Diagnostic Drugs LLC, St Petersburg, Russia) and 4,6-diamino-2phenyl indole (DAPI) for nuclei labeling. Cells were imaged using an LSM710/NLO confocal microscope (Carl Zeiss, Germany). Magnification ×4000. (**a**) Representative confocal images of infected HeLa cell nuclei labeled with DAPI (blue labeling) in fluorescent FDAPI-channel mode. (**b**) Representative confocal images of adenovirus localization (green labeling) in infected HeLa cells. HeLa cells were stained using a diagnostic kit with fluorescent FITC-labeled antibodies to HAdV (Enterprise for the Production of Diagnostic Drugs LLC, St Petersburg, Russia), in fluorescent FITC-channel mode; (**c**) Representative confocal images of infected HeLa cells in phase contrast mode. (**d**) Representative confocal images of adenovirus localization (green labeling) in infected HeLa cells with DAPI-stained nuclei (blue labeling). HeLa cells were stained using a diagnostic kit with fluorescent FITC-labeled antibodies to HAdV (Enterprise for the Production of Diagnostic Drugs LLC, St Petersburg, Russia) in merge mode.

**Figure 5 viruses-16-00386-f005:**
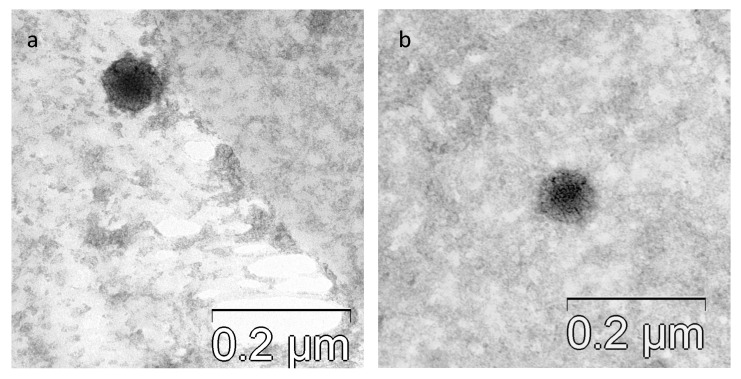
(**a**,**b**) Virions of human adenovirus (strain HAdVC/Novosibirsk/7.134V/2019). Morphology is typical for adenovirus: icosahedral non-enveloped capsid (diameter 70–90 nm). Scale Bar 200 nm.

**Figure 6 viruses-16-00386-f006:**
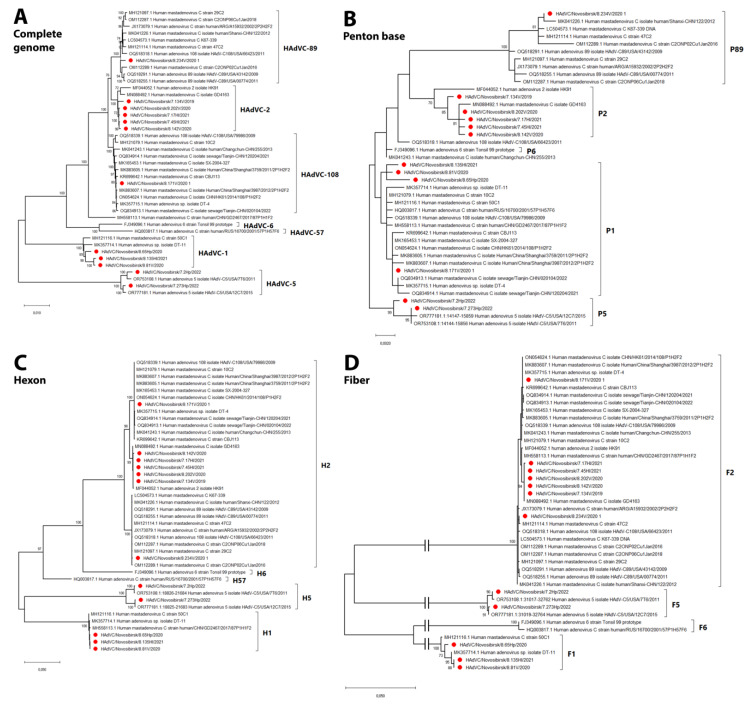
Phylogenetic relationship between our human adenovirus C genotypes and other representative genotypes based on complete genome (**A**), penton base (**B**), hexon (**C**), and fiber (**D**). Maximum likelihood trees were constructed using the best-fit nucleotide substitution model, GTR + G + I (complete genome), K2 + G (penton), TN93 + G (hexon), and HKY + G (fiber) in MEGA X. Our sequences are indicated in red.

**Figure 7 viruses-16-00386-f007:**
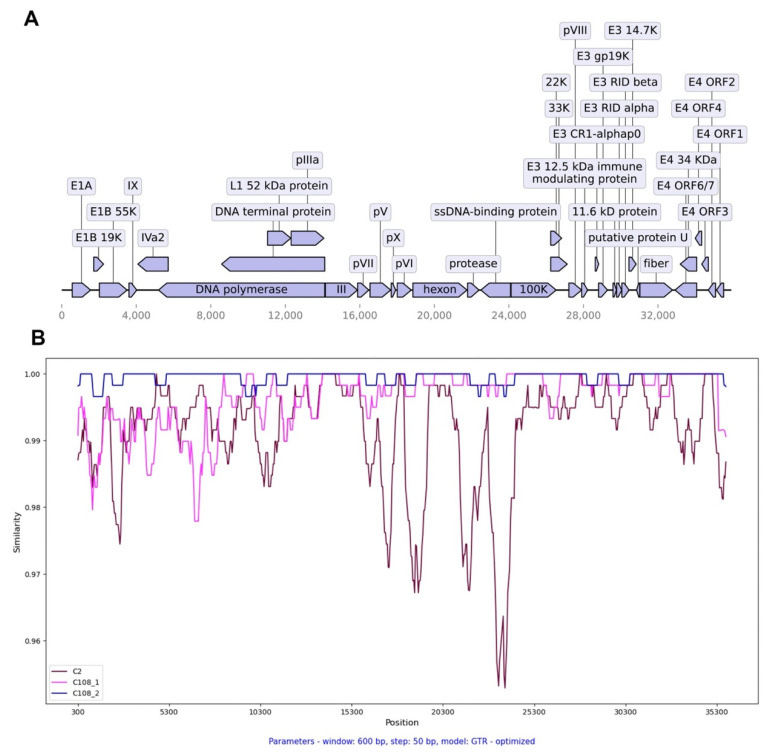
Genome visualization and SimPlot analysis of the HAdVC/Novosibirsk/8.171V/2020 (C108) strain. (**A**)—Genome annotation; (**B**)—SimPlot analysis of 8.171V strain sequence with reference strain sequence (C108_1, OQ108498) and BLAST-hit sequence with the highest e-value (C108_2, ON054624).

**Table 1 viruses-16-00386-t001:** Data on acute respiratory viruses collected from patients aged 0–17 years hospitalized during the period 2019–2022.

	2019–2020			2020–2021			2021–2022		
**Total number of collected samples**	1088			1130			972		
	Age, years (n)								
	0–2	3–6	7–17	0–2	3–6	7–17	0–2	3–6	7–17
	593	298	197	648	336	146	595	218	159
**PCR-positive ***	788			691			616		
	Age, years (n)								
	0–2	3–6	7–17	0–2	3–6	7–17	0–2	3–6	7–17
	445	223	120	401	220	70	402	137	77
**Co-infection**	90								
	Age, years (n)								
	0–2	3–6	7–17	0–2	3–6	7–17	0–2	3–6	7–17
	63	24	3	75	56	5	37	13	1
**Type of virus**	Age, years, n (%)								
	0–2	3–6	7–17	0–2	3–6	7–17	0–2	3–6	7–17
**HAdV**	26 (4.4)	14 (4.7)	3 (1.5)	27 (4.2)	11 (3.3)	2 (1.4)	29 (4.9)	12 (5.5)	3 (1.9)
**HIFV**	122 (20.6)	110 (36.9)	80 (40.6)	0 (0)	0 (0)	0 (0)	37 (6.2)	33 (15.1)	36 (22.6)
**HRSV**	164 (27.6)	58 (19.5)	8 (4.1)	1 (0.2)	2 (0.6)	0 (0)	149 (25)	39 (17.9)	9 (5.7)
**HCoV**	19 (3.2)	15 (5.0)	1 (0.5)	105 (16.2)	36 (10.7)	9 (6.2)	16 (2.7)	3 (1.4)	0 (0)
**HPIV**	67 (11.3)	25 (8.4)	12 (6.1)	58 (8.9)	44 (13.1)	12 (8.2)	36 (6.1)	10 (4.6)	0 (0)
**HMPV**	6 (1.0)	10 (3.3)	2 (1.0)	163 (25.2)	132 (39.3)	30 (20.5)	0 (0)	0 (0)	0 (0)
**HRV**	62 (10.4)	17 (5.7)	8 (4.1)	109 (16.8)	54 (16.1)	21 (14.4)	66 (11.1)	31 (14.2)	7 (4.4)
**HBoV**	47 (7.9)	11 (3.7)	6 (3.0)	36 (5.5)	8 (2.4)	3 (2.1)	17 (2.8)	2 (0.9)	0 (0)
**SARS-CoV-2**	-	-	-	2 (0.3)	1 (0.3)	0 (0)	92 (15.5)	24 (11)	21 (13.2)

Abbreviations: HAdV—adenovirus; HIFV—influenza A and B viruses; HRSV—respiratory syncytial virus; HRV—rhinovirus; HPIV—parainfluenza virus types 1–4; HCoV—alphacoronaviruses (NL63/229E) and betacoronaviruses (OC43/HKU1); HMPV—metapneumovirus; HBoV—bocavirus and SARS-CoV-2—severe acute respiratory syndrome coronavirus 2. All information is presented as numeric and percentage data of medical records analyzed of patients who tested positive for acute respiratory infections during the period 2019–2022. *—At least one of the respiratory viruses HAdV, HIFV, HRSV, HCoV, HPIV, HMPV, HRV, HBoV, or SARS-CoV-2 was identified.

**Table 2 viruses-16-00386-t002:** Pairwise distances of full-genomic nucleotide sequences of human adenovirus obtained in the period 2019–2022.

Group N	N	Type and Genotype	Strain Name	1	2	3	4	5	6	7	8	9	10	11
**1**	**1**	**C5**	HAdVC/Novosibirsk/7.2Hp/2022											
**2**	HAdVC/Novosibirsk/7.273Hp/2022	0.005900										
**2**	**3**	**C2**	HAdVC/Novosibirsk/7.17Hl/2021	0.049293	0.049270									
**4**	HAdVC/Novosibirsk/7.45Hl/2021	0.049221	0.049199	0.000641								
**5**	HAdVC/Novosibirsk/8.202V/2020	0.049288	0.049265	0.000752	0.000613							
**6**	HAdVC/Novosibirsk/8.142V/2020	0.049532	0.049511	0.000975	0.000696	0.000975						
**7**	**C89**	HAdVC/Novosibirsk/8.234V/2020	0.050587	0.050810	0.008064	0.007981	0.008035	0.008288					
**8**	**C108**	HAdVC/Novosibirsk/8.171V/2020	0.049956	0.049963	0.007020	0.006910	0.006964	0.007216	0.008906				
**9**	**C2**	HAdVC/Novosibirsk/7.134V/2019	0.050700	0.050113	0.002788	0.002677	0.002732	0.003011	0.009359	0.008286			
**3**	**10**	**C1**	HAdVC/Novosibirsk/8.135Hl/2021	0.050934	0.050255	0.048392	0.048230	0.048357	0.048572	0.048539	0.045787	0.047484		
**11**	HAdVC/Novosibirsk/8.81V/2020	0.050758	0.050081	0.048243	0.048077	0.048211	0.048419	0.048449	0.045575	0.047338	0.001056	
**12**	HAdVC/Novosibirsk/8.65Hp/2020	0.049590	0.048825	0.047179	0.047045	0.047174	0.047294	0.047413	0.044300	0.046720	0.005634	0.005410

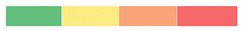
 0.001–0.05

**Table 3 viruses-16-00386-t003:** Molecular biological and virologic data about human adenoviruses C collected during the period 2019–2022 in Novosibirsk, Russia.

N	Strain	GenBank	Genetic Characteristics					Viral Titers, lgTCID_50_/mL
			Classification by				Sequence Length, n.p.	
			Fiber Gene	Hexon Gene	Penton Gene	Complete Genome		
1	HAdVC/Novosibirsk/8.65Hp/2020	MZ151860	C1	C1	C1	C1	35,984	3.75 ± 0.21
2	HAdVC/Novosibirsk/8.81V/2020	MZ151866	C1	C1	C1	C1	35,976	3.50 ± 0.23
3	HAdVC/Novosibirsk/8.135Hl/2021	ON152653	C1	C1	C1	C1	36,009	3.45 ± 0.21
4	HAdVC/Novosibirsk/8.171V/2020	MZ151865.1	C2	C2	C1	C108	35,913	3.50 ± 0.23
5	HAdVC/Novosibirsk/8.142V/2020	MZ151864.1	C2	C2	C2	C2	35,928	3.75 ± 0.21
6	HAdVC/Novosibirsk/7.134V/2019	MZ151861	C2	C2	C2	C2	35,921	3.75 ± 0.21
7	HAdVC/Novosibirsk/8.202V/2020	MZ151863.1	C2	C2	C2	C2	35,909	3.25 ± 0.21
8	HAdVC/Novosibirsk/8.234V/2020	MZ151862.1	C2	C2	C89	C89	35,907	3.75 ± 0.21
9	HAdVC/Novosibirsk/7.17Hl/2021	ON152651	C2	C2	C2	C2	35,934	3.75 ± 0.21
10	HAdVC/Novosibirsk/7.45Hl/2021	ON152652	C2	C2	C2	C2	35,920	3.75 ± 0.21
11	HAdVC/Novosibirsk/7.2Hp/2022	ON152649	C5	C5	C5	C5	35,910	3.40 ± 0.31
12	HAdVC/Novosibirsk/7.273Hp/2022	ON152650	C5	C5	C5	C5	35,915	3.30 ± 0.35

Note: genotype C1 is colored in pale pink 
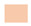
, genotype C2 is colored in yellow 
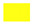
, genotype C5 is colored in green 
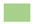
, genotype C89 is colored in brown 
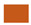
, and genotype C108 is colored in red 

.

## Data Availability

The original contributions presented in the study are included in the article/Supplementary Material, further inquiries can be directed to the corresponding author.

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
