# Peer review of "Genetic Diversity of the Human Adenovirus C Isolated from Hospitalized Children in Russia (2019–2022)"

_viruses, 2024, doi:10.3390/v16030386_

Round 1
Reviewer 1 Report
Comments and Suggestions for Authors
This paper by Kurskaya and colleagues reports the characterization by whole genome sequencing of 12 HAdV isolates of species HAdV-C.
The use of idiomatic English language in the manuscript requires revision.
Reference 2 needs to be updated to a more recent edition of Fields Virology.
Based on the reported numbers, 127 out of 3190 samples tested positive for adenovirus. It is unclear why only 12 of these samples were moved forward for whole genome sequencing. Were there any attempts made to type the other 115 detected HAdVs? Amplification and sequencing of the hexon coding region is a viable option. Were all 127 specimens put in culture to attempt viral isolation? Without this information the data on detailed typing of the 12 HAdV-C viruses is out of context as the authors do not provide information about how many of the 115 detected HAdVs belong to species C.
The data and figures (Figs 1, 2, 3 and 4) illustrating cytopathic effect in various cell lines, IF and EM findings are not necessary since these are standard procedures in clinical virology. A brief description in the Methods section will suffice.
In Table 2, it would be highly desirable to include the sequences of the prototype strains of genotypes C1, C2, C5, C6, C57, C89, C104 and C108. The comparison of the characterized strains is not descriptive enough.
The use of Simplot analysis is highly recommended to help the reader visualize the differences between the characterized virus genome and the most closely related genotypes of species C.
As indicated by the asterisks in table 3, some of the identified genomes may be novel intertypic recombinants and therefore this reviewer wants to encourage the authors to submit genomic analysis data to the human adenovirus working group (http://hadvwg.gmu.edu) for evaluation of candidate new genotypes.
With such limited number of detected HAdVs typed, the conclusion that HAdV-C2 was the predominant genotype is not supported.
Comments on the Quality of English LanguageThe use of language requires extensive revision to correct incorrect use of terminology (example: "fibrillar" on line 51), grammatical errors, spelling, etc.
The first paragraph of the introduction needs work. More detail on the classification criteria needs to be provided. 114 genotypes have been described to the present.
Author Response
Dear Reviewer #1,
Thank you for your useful comments and questions that helped to improve the manuscript. Please see below each point for the detailed response on how we incorporated your feedback into the manuscript.
This paper by Kurskaya and colleagues reports the characterization by whole genome sequencing of 12 HAdV isolates of species HAdV-C.
The use of idiomatic English language in the manuscript requires revision.
We agree with reviewer’s assessment and tried to make proofreading.
Reference 2 needs to be updated to a more recent edition of Fields Virology.
We have changed the references and new information/
[2] ICTV. International Committee on Taxonomy of Viruses. Available at: http://www.ictvonline.org/virusTaxonomy.asp. Accessed July 25, 2023.
[3] Human Adenovirus Working Group (2024). http://hadvwg.gmu.edu. (Accessed February 1, 2024)
Based on the reported numbers, 127 out of 3190 samples tested positive for adenovirus. It is unclear why only 12 of these samples were moved forward for whole genome sequencing. Were there any attempts made to type the other 115 detected HAdVs? Amplification and sequencing of the hexon coding region is a viable option. Were all 127 specimens put in culture to attempt viral isolation? Without this information the data on detailed typing of the 12 HAdV-C viruses is out of context as the authors do not provide information about how many of the 115 detected HAdVs belong to species C.
We randomly selected 30 samples out of 127 PCR positives with the highest Ct values (18-23), representing different ages and sexes, which we attempted to isolate in cell cultures.
The data and figures (Figs 1, 2, 3 and 4) illustrating cytopathic effect in various cell lines, IF and EM findings are not necessary since these are standard procedures in clinical virology. A brief description in the Methods section will suffice.
We preferred to add information about the dedicated strain HAdVC/Novosibirsk/7.134V/2019.
In Table 2, it would be highly desirable to include the sequences of the prototype strains of genotypes C1, C2, C5, C6, C57, C89, C104 and C108. The comparison of the characterized strains is not descriptive enough.
By using the additional method that we have done for improving our results – the Simplot analysis - we have found the fifth genotype C108, and therefore have rewritten the Text. We have included Table 3 with prototype strains nucleotide sequences pairwise identities comparison. Kindly, find new information in the Article (Fig. 6, Table 2 & 3, Fig. 7, Suppl. Table S1). Thus, no non-described recombinations in penton base, hexon and fiber ORFs among studied strains were found.
The use of Simplot analysis is highly recommended to help the reader visualize the differences between the characterized virus genome and the most closely related genotypes of species C.
As indicated by the asterisks in table 3, some of the identified genomes may be novel intertypic recombinants and therefore this reviewer wants to encourage the authors to submit genomic analysis data to the human adenovirus working group (http://hadvwg.gmu.edu) for evaluation of candidate new genotypes.
With such limited number of detected HAdVs typed, the conclusion that HAdV-C2 was the predominant genotype is not supported.
We agree with Reviewer and tone down the phrase.
The use of language requires extensive revision to correct incorrect use of terminology (example: "fibrillar" on line 51), grammatical errors, spelling, etc.
We agree with reviewer’s assessment and tried to make proofreading.
The first paragraph of the introduction needs work. More detail on the classification criteria needs to be provided. 114 genotypes have been described to the present.
We`ve updated the information about genotypes and rewritten the introduction almost in general - `.. The human adenovirus (HAdV) has double-stranded DNA and a non-enveloped icosahedral structure which includes three viral capsid proteins; hexon, penton base, and fiber [1]. HAdV belongs to the genus Mastadenovirus (family Adenoviridae) [2]. Currently, seven different species of HAdV (from A to G) have been identified [2] while, according to the classification criteria of the Human Adenovirus Working Group, 114 HAdV genotypes have been assigned [3].
Adenovirus infection in children may cause respiratory illnesses (bronchitis, laryngotracheitis, tracheitis, rhinopharyngitis, pharyngitis, pneumonia), or other disease complications (otitis, conjunctivitis, gastrointestinal symptoms, hemorrhagic cystitis, mesenteric adenitis, myocarditis, encephalitis) [4,5,6,7]. It is believed that among all cases of acute respiratory viral infection (ARVI) in children, HAdV accounts for at least 5-10% [8], and such species as HAdV-C (namely, genotypes C1, 2, 5, 6, 57, 89, 104, 108) may be associated with local ARVI outbreaks among children [9, 10, 11]. In Russia, the incidence of HAdV was 6.9% in children aged 0 - 17 years during the period 2004-2014 with co-infections occurring in 41% of cases [12]. Among all types of adenovirus, the proportion of HAdV-C in children under one year old was 60%, in preschoolers (1-7 years old) — 53.6%, and in school children (7-17 years old) — 21.4% [12]. According to Yatsishina et al. [12], in Russia during the period 2004-2014, the most common HAdV etiological agents among children were HAdV-C2 (20.9%) and HAdV-C6 (11.6%), the average age with adenovirus infection in children being 3 years and 11 months.
In Russia currently, knowledge of the epidemiology and genotyping of HAdV-C infection is lacking. Our results present the virological, epidemiological and genotyping data of HAdV-C arising from a retrospective observational study that collected laboratory-confirmed HAdV cases in the period 2019-2022, at the two Novosibirsk Children’s Municipal Clinical Hospitals №6 and №3. Here, we report the molecular epidemiological and virological characteristics of HAdV with C1, C2, C5 and C89 and C108 detected in Russia among 3,109 samples from children 0-17 years old. Overall, the study sheds light on the epidemiological situation and viral characteristics of HAdV-C and its circulating genotypes in one region of Russia.
The aim of the study was the molecular epidemiological and virological analysis of HAdV with C1, C2, C5, C89, and 108 detected among 3,190 samples from children 0-17 years old. Overall, the study sheds light on epidemiological situation, viral characteristics of HAdV-C and its circulating genotypes in one region of Russia.`.
Finally, we corrected the text and try to fix all the comments. Please see it again.

Reviewer 2 Report
Comments and Suggestions for Authors
Please see attached.

There are understandably a few areas where the English is a bit choppy -- please see my comments by line item, but these are easy to remedy.
Author Response
COMMENTS FOR THE AUTHORS:
Reviewer #2: This report summarizes a 4-year longitudinal (2019-2022) study of viral pathogens detected at 2 pediatric hospitals within Novosibirsk, Russia with an emphasis on the genetic characterization of 12 human Adenovirus C (HAdV C) genotypes detected. The authors have done an extensive virologic workup of these HAdV C viruses including traditional methods (virus isolation and electron microscopy) and advanced technologies (qPCR, next generation sequencing, and phylogenetic analysis). The authors include important and useful surveillance data over the 4 years examined on other respiratory viral pathogens, categorized by patient age, in a summary table (Table 1). Although the authors include brief descriptions of co-infections, they are not shown in tabular or other format – I would find it useful for the co-infection data to be shown, but if data will not be shown then the authors should indicate that with the phrase “data not shown”.
The phylogenetic analysis of HAdV C viruses showed very interesting patterns – there were fully intact C1 and C2 genotypes circulating, based on examination of the separate hexon, penton, and fiber genes and the whole genome. But the analysis also showed some promiscuity or recombination of the penton gene between C1 and C2 to yield an undescribed HAdV C genotype (Table 3 N4), between C2 and C89 yielding a C89 genotype based on the whole genome (Table 3 N8), and between C5 and an undescribed HAdV C genotype (Table 3 N11 and N12). The authors discuss C89 and circulation in China, presumably implying that this may be a possible origin of C89 in Novosibirsk – but do not state this outright.
Can the authors elaborate on these observations that only the penton seemed to be involved in putative recombination events (not the hexon or fiber) and why might this be? Overall, this is an important documentation of circulating viral pathogens in a hospitalized pediatric population over 3 respiratory seasons in Novosibirsk, Russia, and should be published to share these data with the world. I do have several comments/suggestions as indicated below by line or Table. Line 22: instead of "few" maybe replace with "rarely reported"?
Dear Reviewer #2,
Thank you for your useful comments and questions that helped to improve the manuscript. Please see below each point for the detailed response on how we incorporated your feedback into the manuscript.
Line 33: replace "registered" with "recorded"
The correction was done.
Line 36: May want to mention that your analysis of the HAdV C viruses indicated signs recombination in the isolates examined
By using the additional method that we have done for improving our results – the Simplot analysis - we have assigned to C108 genotype strain HAdVC/Novosibirsk/8.171V/2020, and therefore have rewritten the Text. Kindly, find new information in the Article (Fig. 6, Table 2 & 3, Fig. 7, Suppl. Table S1). Thus, no recombinations among studied strains were found.
Line 44: "genus" is the proper word to use here as "genera" refers to the plural of genus
The correction was done.
Line 46: please include the reference supporting the statement that there are 113 HAdV types. I found a reference showing 111 HAdVt types at this link http://hadvwg.gmu.edu/ and numerous references to more than 100 types
We have added the missed reference (doi: 10.3389/fmicb.2022.911694).
Line 50: suggest re-phrasing as follows: "... corners and each penton bears a fiber."
The correction was done
Line 51: "genus" is the proper word to use here as "genera" refers to the plural of genus.
The correction was done
Line 51: change "proteins" to "protein" or delete the word entirely.
The correction was done.
Line 51: change "fibril" to "fiber".
The correction was done.
Line 54: remove "are".
The correction was done.
Line 63: replace "reason of" with "associated with".
The correction was done.
Line 66-67: 60% and 53.6% add up to more than 100%.
We`ve paraphrased it to “In Russia, the incidence of HAdV was 6.9% in children aged 0 - 17 years during the period 2004-2014 with co-infections occurring in 41% of cases [12]. Among all types of adenovirus, the proportion of HAdV-C in children under one year old was 60%, in preschoolers (1-7 years old) — 53.6%, and in school children (7-17 years old) — 21.4% [12].
Line 69: "all" should be spelled as "al."
The correction was done.
Line 70: shouldn't this read as "the most common HAdV" NOT "ARVI"?
We agree with Reviewer and have done the correction.
Line 84: delete the word "would"
The correction was done.
Line 89: this would be more accurate to say "...in an Oblast of Russia." rather than as written referring to all of Russia.
The correction was done.
Lines 111 - 119: The authors should indicate when real-time reverse transcription PCR was employed (for the RNA-based viruses) to distinguish from real-time PCR (for DNA viruses).
We have added information - `...Viral nucleic acids were extracted from all clinical samples using RNA/DNA extraction kit «RIBO-sorb» (Interlabservice, Russia) according to the manufacturer’s instructions. The reverse transcription of extracted viral nucleic acids was immediately performed using commercial kit "REVERTA-L" (Interlabservice, Russia)…`
Line 118: Were enteroviruses included as part of the rhinovirus analysis
No, we did not conduct a separate analysis for the detection of enteroviruses, For the screening of respiratory viruses, we used a commercial PCR-kit that detects all rhinoviruses and some genetically related enteroviruses (as indicated in the manufacturer's instructions).
Line 128: add a comma after PCR.
The correction was done.
Line 130: daily what? – we have added missed words.
The “daily monolayer of”
Lines 175 -183: Why not list these in sequential order?
We have written the list in sequential order.
Line 187: two words "software" "was"
The correction was done.
Line 190: Is this a conventional model? I am not familiar with this, so have no input.
Yes, this is the standard description of phylogenetic tree reconstruction. Actually, when constructing the tree using MEGA soft you should choose the best fit nucleotide substitution model. It can be done automatically by MEGA program that takes into account different parameters when you apply special tool inside. Then you just use and mention the details which the soft give you. In our study the best models were different for different genes. We modified the description.
Line 203: "ARVI" is used elsewhere in the report -- please be consistent with what is used.
Here we wrote about “ARI”, because not all of 3,190 children had an actually “viral” infection of the respiratory tract (some of them had a bacterial infection). Next, we clarified that 74.4% had at least one virus (HAdV, HIFV, HRSV, HCoV, HPIV, HMPV, HRV, HBoV, or SARS-CoV-2) detected in respiratory tract, and it is mean that they were suffered with “ARVI”.
Lines 212-215: what do these denominators represent (1836 and 502)? these values should be shown on the table for convenience to the reader.
These denominators (1836 and 502) represent the total number of samples collected from patients 0 – 2 and 7 – 17 years old, respectively.
Line 212: should the "1248" actually be "1428" for all aged 0-2 with viral infection detected? this would change % to 78%.
We think the total percentage of viruses cannot be summarized as indicated in the comment, because in some cases these viruses occur as a coinfection in one patient (in one sample).
Line 213: "least susceptible" I suggest deleting this phrase and just say school-aged cohort showed least percent/number of viral infections.
The correction was done.
Line 215: insert "older than" to replace "elder"
The correction was done.
Lines 216-217: co-infection data do not appear to be shown in a table or other manner anywhere -- so should say "data not shown" or show the data.
We have added information about co-infection data into Table 1 and made new Figure 1. Kindly find it in the Text.
Lines 224-225: co-infection data do not appear to be shown in a table or other manner (same comment as above).
We have added information about co-infection data into Table 1. Kindly find it in the text.
Table 1: it would be best to include totals for each age group/season relating to viruses detected, e.g., for 0-2yrs age total of samples where a single virus was detected equals 504 with corresponding % of specimens collected being 85% (504/593).
We have added the total number of PCR-positive samples and the number of co-infections in each age group/season in the table 1.
Line 237: should "registration" be replaced with "recorded"?
The correction was done.
Figure 3: The letters seem to be misplaced on this figure...does the Note (Lines 265-285) below refer to Figure 3?
Thank you very much! Indeed, the letters were misplaced on this Figure. We tried to fix them.
Yes, the Note (Lines 265-285) below refer to Figure 4 (in previous version Fig.3).
Line 296: Should refer to these with C species designation: C1, C2, etc.
The corrections were done. We have specified information as - C1, C2, C5, C89, and C108.
Line 298: add "and" after "hexon gene,".
The corrections were done.
Line 299: replace "clearly showed belonging to the " with "of".
The corrections were done.
Line 311: which strains? the C5 strains? Please indicate with strains here for clarity. Suggest referring to figure or table as well for clarity.
We`ve made the referring in the Text - Fig. 6; Table 2, 3; Suppl. Table S1.
Line 313: delete "that"
The correction was done.
Line 317: delete "By the way, "
The correction was done.
Lines 318-319: delete "At the same time"
We have corrected two sentences – “…This strain had substitutions characteristic of genotype 89 (A363E, L154Q), while at the same time the strain had fiber and hexon genes belonging to the C2 genotype…”
Line 319: delete "belonging"
The same correction as above.
Lines 319-320: Reword this sentence for clarity -- what is trying to be conveyed here?
The same correction as above.
Line 324: suggest referring to the cell within Table 3 rather than "marked white", something like this: “(Table 3, N4)”
The correction was done.
Line 326: delete "further" and "singed"
The correction was done.
Line 331: change "consider" to "considering"
The correction was done.
Table 2 Note on page 13: Can authors add explanation for the heat map and what it represents? Line 372: change "was" after "years old" to "were"
We agree with the comment – this Note does not have any additional content, which is different from the text of chapter. We have removed this. As we`ve written in the Text: ` …In general, the strains of human adenovirus C studied formed several groups according to the pairwise distances of full-genomic nucleotide sequences, as confirmed by the similarity of genotypes 1, 2, 5, 89, and 108 (Table 2; Suppl. Table S1)...` This information support the fact that we can see different group according to the these genetic distances.
Line 373: I think this number should read as 1428 not 1248
We think the total number of viruses cannot be summarized as indicated in the comment, because in some cases these viruses occur as a coinfection in one patient (in one sample).
Line 379: "occupy lower rate followed by HIFV," this contradicts your statement above how HIFV were most common agent in 2019-2020...
We agree with the comment and have rewritten the phrase: “…In toto, as shown in our previous work, the assessment reports about etiological agents of ARVI during the period 2019-2022 that HAdV presented at a lower rate, following after HIFV, SARS-CoV-2, HRSV, and HMPV].”
Line 382: If you are comparing how many cases of HAdV from all HAdV cases were for the group less that 2yrs old, shouldn't the denominator be total number of HAdV cases and NOT total samples from < 2yrs old? So therefore, the ratio would be 82/127 or 65%...
We did not consider the prevalence of adenoviruses detected among the certain age from the total number of HAdV cases, since the sample size of each age group differed. Therefore, we calculated the prevalence of adenoviruses in each age group.
Line 386-388: These co-infection data were not shown -- so if this is important, these data should be shown. I think a table showing the detected co-infections would add to the scientific knowledge, so I suggest adding this.
We have added information about co-infection data into Table 1. Kindly find it in the text.
Line 387: suggest re-wording something like this: HAdV were detected in triple infections as follows...
The corrections were done in Results and Discussion parts.
Line 391: change "genotype" to HAdV species C.
The correction was done.
Line 395: add "species C" after "HAdVs".
The correction was done.
Finally, we corrected the text and try to fix all the comments. Please see it again.

Reviewer 3 Report
Comments and Suggestions for Authors
Introduction:
The introduction contains a lot of information, but it could benefit from several improvements and corrections:
· The organization of information is scattered, leading to a lack of coherence. It's important to structure the introduction in a way that flows logically from one point to another.
· The statement regarding the number of species of HAdV needs clarification. While there are indeed multiple types classified under different species, the exact number and classification might differ based on ongoing research. Ensure the accuracy of the information presented.
· Statements like "wide variety of known HAdVs" without specifying the extent or significance of this diversity lack depth and clarity. Elaborating on this diversity could enhance the depth of understanding.
· The description of the HAdV capsid structure might need verification for precision, especially the number and distribution of structural units.
· Some statistics provided lack specific sources or dates. Ensure all statistical data and references are current and come from reliable sources.
· The epidemiological data lacks specificity regarding geographical locations, making it challenging to understand the relevance and application of the information.
· While it discusses the prevalence of HAdV in Russia, it lacks broader context or comparisons with global trends, which would provide a more comprehensive understanding of the virus's impact.
· The language used needs refinement for clarity. Some sentences are complex and could be simplified for easier comprehension.
· The admission of lacking knowledge about the epidemiology and genotyping of HAdV-C in Russia might benefit from rephrasing to indicate a need for further research rather than complete lack of knowledge.
· The objectives of the study are outlined but are quite broad. It might be beneficial to specify the research questions or aims more clearly.
Materials and methods:
· While it mentions the periods of sample collection, additional information about the number of samples collected, demographics, and inclusion/exclusion criteria would add depth to the methodology.
· Provide more specific details about the methodology for virus isolation on different cell lines. Mention specific media used, incubation periods, and criteria for confirming viral growth.
· Elaborate on the staining and imaging techniques used for the evaluation of cytopathogenic effects (CPE) and immunofluorescence assays. Describe the criteria for identifying HAdV in infected cells more explicitly.
· While it describes various techniques used, some steps lack specific details. For instance, the procedures for sample collection, RNA/DNA extraction, and sequencing could be more elaborately described to guide others in replicating the study.
· There are a few typos and technical errors present, such as missing spaces between sentences and numbering errors, like "Novem101 ber-April." Careful proofreading and editing would improve the section's professionalism.
· Details regarding positive and negative controls are mentioned but not described in detail. Including how these controls were chosen and their role in ensuring the accuracy of the results would strengthen the methodology.
· While the RT-PCR kits used are mentioned, details about the specific primers, probes, or target genes used for each virus detection would be valuable for reproducibility.
· The section on sequence analysis lacks clarity in explaining why specific software or models were chosen. Elaborating on the rationale behind the chosen methods would enhance the scientific rigor of the analysis.
· The statistical analysis part lacks detail on the variables analyzed and the specific tests conducted beyond the Chi-square test. Providing more context on the approach used and the rationale for the statistical tests would strengthen this section.
Results:
This section presents valuable findings but could benefit from certain improvements and clarifications:
· The sampling method and representativeness of the studied population should be clarified to ensure the findings accurately reflect the broader population. Details on any bias in sample collection and its potential impact on the results would be insightful.
· A more detailed discussion about the clinical implications of co-infections and the potential severity of infections compared to single viral infections would enrich the interpretation.
· While the identification methods (PCR, immunofluorescence, electron microscopy) were described, providing details about the sensitivity and specificity of these techniques could enhance the understanding of the identified viruses.
· Explaining the clinical significance of the observed genomic variations within strains and how these variations might impact disease severity or treatment would add depth to the findings.
· Exploring the implications of similar virus titers across different genotypes on disease severity or transmission rates would be informative.
Discussion:
· The discussion lacks a clear introductory statement summarizing the objectives or the purpose of the study. The context of the study regarding the significance of understanding HAdV's epidemiology in acute respiratory infections requires expansion to provide a broader understanding.
· The structure is disjointed, lacking clear subsections to segregate different aspects like epidemiological observations, virological characteristics, genomic analyses, and their implications.
· Points regarding the incidence among different age groups and the prevalence of various viruses could be presented more systematically for better comprehension.
· The discussion frequently refers to earlier works (references [15, 16, 17]) without specifying the findings or context, making it hard to follow without referring to those sources directly.
· Data, such as the incidence rates among different periods and age groups, lack precise statistical details or comparative analyses.
· The discussion lacks deeper analysis or interpretation of the observed trends. It presents data without adequately discussing the potential reasons behind variations in virus prevalence among different periods or age groups.
· Further analysis of the observed co-infection patterns or potential implications of co-infections on disease severity or clinical outcomes is absent.
· The discussion lacks clarity regarding the significance of lower viral activity observed in comparison to other studies, requiring a more detailed explanation or possible implications of this difference.
· The discussion of the genetic variations and phylogenetic analyses seems disconnected from the main narrative, making it hard to understand how these findings relate to the broader context of the study.
· The findings regarding the unassigned strain (HAdVC/Novosibirsk/8.171V/2020) require a more detailed discussion regarding its implications and potential causes for the lack of classification.
· The conclusion is abrupt and lacks a summary of key findings. It should reiterate the most significant outcomes and their implications for the field of study.
· The conclusion regarding the prevalence of genotypes lacks deeper insights or implications for future research or clinical practice.
· The conclusion could be strengthened by suggesting future research directions, such as exploring the clinical significance of different genotypes, understanding the implications of co-infections, or further investigating unassigned strains.
· The study's significant contributions, especially the identification of the HAdV-C89 genotype in Russia for the first time, could be emphasized more explicitly in the discussion and conclusion.
Comments on the Quality of English LanguageExtensive editing of English language required
Author Response
Dear Reviewer #3,
Thank you for your useful comments and questions that helped to improve the manuscript. Please see below each point for the detailed response on how we incorporated your feedback into the manuscript.
Introduction:
The introduction contains a lot of information, but it could benefit from several improvements and corrections:
The organization of information is scattered, leading to a lack of coherence. It's important to structure the introduction in a way that flows logically from one point to another.
The statement regarding the number of species of HAdV needs clarification. While there are indeed multiple types classified under different species, the exact number and classification might differ based on ongoing research. Ensure the accuracy of the information presented.
We have upgraded the information according to the Human Adenovirus Working Group and rewrote the sentence – “…Currently, seven different species of HAdV (from A to G) have been identified [2] while, according to the classification criteria of the Human Adenovirus Working Group, 114 HAdV genotypes have been assigned [3]…`. Additionally, we have supported this sentence by the new reference – 3 - Human Adenovirus Working Group (2024). http://hadvwg.gmu.edu. (Accessed February 1, 2024).
Statements like "wide variety of known HAdVs" without specifying the extent or significance of this diversity lack depth and clarity. Elaborating on this diversity could enhance the depth of understanding.
We agree with Reviewer and removed this phrase.
The description of the HAdV capsid structure might need verification for precision, especially the number and distribution of structural units.
We agree with Reviewer and have rewritten the introduction in general.
Some statistics provided lack specific sources or dates. Ensure all statistical data and references are current and come from reliable sources.
We have taken into account your comment and expanded the paragraph describing the statistical analysis - `… Descriptive statistics included mean and standard error (for quantitative traits), absolute and relative proportion (for qualitative traits). Categorial data were analyzed using a two-tailed Chi-square test undertaken to compare infection rates for respiratory viruses among different age groups. …`.
However, with regard to statistical tests, we used only Chi-squared, given the type of our data.
The epidemiological data lacks specificity regarding geographical locations, making it challenging to understand the relevance and application of the information.
Novosibirsk is a large city located in the southwestern part of Siberia and is one of the principal cities of the Asian region of Russia, considered to be the third largest city in the country after Moscow and Saint Petersburg. In our study the samples were taken in two hospitals of Novosibirsk (Children’s Municipal Clinical Hospital №6, and №3). So, in the part of Materials and Methods we have mentioned only the town.
While it discusses the prevalence of HAdV in Russia, it lacks broader context or comparisons with global trends, which would provide a more comprehensive understanding of the virus's impact.
Despite the fact that we cannot support the conclusions about the genotypes, prevalent in our study, because of the limited number of genotyped samples, we showed evidence of the circulation of various HAdV variants. For the first time the circulation of the HAdV-C89 and C108 genotypes in Russia was shown, according to the results of phylogenetic analysis based on penton nucleotide sequences and complete genomes.
The language used needs refinement for clarity. Some sentences are complex and could be simplified for easier comprehension.
We agree with reviewer’s assessment and tried to make proofreading.
The admission of lacking knowledge about the epidemiology and genotyping of HAdV-C in Russia might benefit from rephrasing to indicate a need for further research rather than complete lack of knowledge.
We have re-written the sentence into – “…Overall, the study sheds light on epidemiological situation, viral characteristics of HAdV-C and its circulating genotypes in one region of Russia.…”
The objectives of the study are outlined but are quite broad. It might be beneficial to specify the research questions or aims more clearly.
We have re-written these sentences into – “…The aim of the study was the molecular epidemiological and virological analysis of HAdV with C1, C2, C5, C89, and 108 detected among 3,190 samples from children 0-17 years old…`
Materials and methods:
While it mentions the periods of sample collection, additional information about the number of samples collected, demographics, and inclusion/exclusion criteria would add depth to the methodology –
We have added information about exclusion criteria - `…Exclusion criteria for samples were the diseases duration for more than 7 days..`
All other information about number of samples and respiratory symptoms among enrolled children were mentioned in the part of Sample Collection in Material and Methods.
Also, we have added some limitations:
`… Our study sample was obtained only from hospitalized patients. It is also known that the state of virus infection of a carrier can be asymptomatic. Accordingly, the distribution of viruses among the Russian population may differ from our sample results. We also did not analyze social and behavioral factors that may serve as potential drivers of viral interference, and we cannot exclude the possibility of surveillance artefacts. The strength of our data is based on sequential observations from both pre-pandemic and pandemic periods through research spanning three years, and the collection of samples from 3,190 patients 0-17 years old. Annual, correct monitoring of seasonal fluctuations of common respiratory viruses has the potential to improve general knowledge about viral interactions. However, as global surveillance systems both for non-influenza respiratory viruses and for non-COVID-19 research remain limited in both scope and funding, the ability to undertake such investigations remains challenging…`
Provide more specific details about the methodology for virus isolation on different cell lines. Mention specific media used, incubation periods, and criteria for confirming viral growth.
We have added next information - 96-well plates with daily monolayer of HeLa, HEp-2, Vero, SPEV, and Caco-2 cells in growth medium were washed with a Henks solution and then inoculated with 30 μl aliquots of the clinical sample. After 1 h of incubation at 37°C and 5% CO2supernatant was removed and 200μL/well of DMEM (Capricorn Scientific, Germany) with 2% of FBS (Capricorn Scientific, Germany) and 100 μg/mL of gentamicin sulfate (BioloT, Russia) were added. Intact HeLa, HEp-2, Vero, SPEV, and Caco-2 cells were used as controls in each plate. Infected plates were incubated at 37 °C, 5% CO2 up to 7 days and were daily checked under the microscope (Micromed I, Russia) by the presence of a cytopathogenic effect (CPE) manifested by cell death.
Elaborate on the staining and imaging techniques used for the evaluation of cytopathogenic effects (CPE) and immunofluorescence assays. Describe the criteria for identifying HAdV in infected cells more explicitly.
The CPE score was assessed by the presence of cytopathic changes in the cell monolayer, the destruction of the cell monolayer, the formation of clusters of rounded apoptotic cells, loss of intercellular contacts, rounding of the cell shape. Intact HEp-2 and Vero cells were used as controls to compare the morphological state of inoculated and intact cells monolayer.
While it describes various techniques used, some steps lack specific details. For instance, the procedures for sample collection, RNA/DNA extraction, and sequencing could be more elaborately described to guide others in replicating the study.
We have added the information – “…Samples were placed in tubes with transport medium (Dulbecco’s modified Eagle’s medium (Capricorn Scientific, Germany) with 0.5% bovine serum albumin, 100 μg/mL of gentamicin sulfate (BioloT, Russia), and 50 units/mL of amphotericin B (BioloT, Russia)) and were stored at at 2°C to 8°C before analysis but not more than 48 h…’’
There are a few typos and technical errors present, such as missing spaces between sentences and numbering errors, like "Novem101 ber-April." Careful proofreading and editing would improve the section's professionalism.
We`ve tried to make a careful proofreading. Thank you for your comment!
Details regarding positive and negative controls are mentioned but not described in detail. Including how these controls were chosen and their role in ensuring the accuracy of the results would strengthen the methodology.
Each kit of reagents was targeted to a conservative site of the viral genome. Unfortunately, the exact region of genome was not specified in the instructions as the intellectual property of the company.
While the RT-PCR kits used are mentioned, details about the specific primers, probes, or target genes used for each virus detection would be valuable for reproducibility.
We have added some information – `…Viral nucleic acids were extracted from all clinical samples using RNA/DNA extraction kit «RIBO-sorb» (Interlabservice, Russia) according to the manufacturer’s instructions. The reverse transcription of extracted viral nucleic acids was immediately performed using commercial kit "REVERTA-L" (Interlabservice, Russia)…`
The section on sequence analysis lacks clarity in explaining why specific software or models were chosen. Elaborating on the rationale behind the chosen methods would enhance the scientific rigor of the analysis.
The models were chosen automatically as suggested by the soft for different gene target. We have added some information – Phylogenetic trees were built via MEGA 11 using Maximum Likelihood following results of the best-fit model test in MEGA (complete genome sequence – GTR + G + I; penton – T92 + G + I; hexon – TN93 + G; fiber – HKY + I).
The statistical analysis part lacks detail on the variables analyzed and the specific tests conducted beyond the Chi-square test. Providing more context on the approach used and the rationale for the statistical tests would strengthen this section.
We agree with the Reviewer that our research need the expansion of the paragraph describing the statistical analysis. We`ve added next: `…Data analysis was performed using Microsoft Excel 16.16.2 for MacOs and GraphPad 9.1.1 (GraphPad Software). Descriptive statistics included mean and standard error (for quantitative traits), absolute and relative proportion (for qualitative traits). Categorial data were analyzed using a two-tailed Chi-square test undertaken to compare infection rates for respiratory viruses among different age groups...`
However, with regard to statistical tests, we used only Chi-squared, given the type of our data.
Results:
This section presents valuable findings but could benefit from certain improvements and clarifications:
The sampling method and representativeness of the studied population should be clarified to ensure the findings accurately reflect the broader population. Details on any bias in sample collection and its potential impact on the results would be insightful. –
In our study the samples were taken from all patients that got into the hospitals of Novosibirsk (Children’s Municipal Clinical Hospital №6, and №3). Age of patients were 0-17 years. The diseases duration - no more than 7 days
We did not repeat this information in the part of Result, but discussed it below in the part of Discussion.
A more detailed discussion about the clinical implications of co-infections and the potential severity of infections compared to single viral infections would enrich the interpretation.
Because we did not include clinical effects in the purpose of the study, we did not diside to discuss it in this Article. Besides, we have collected just a few samples with confirmed HAdV - only 127, that does not allow us to speculate on the topic about clinical implications.
We agree with the Reviewer that this theme is also interesting for the virologist community, and it affects the area of our interest too. As example, earlier we published the broader analysis of clinical cases on how the SARS-CoV-2 impact on the etiology of viral agents [Changes in the Etiology of Acute Respiratory Infections among Children in Novosibirsk, Russia, between 2019 and 2022: The Impact of the SARS-CoV-2 Virus. Viruses 2023, 15 (4), 934; https://doi.org/10.3390/v15040934].
Also, in the current article we did not note a higher prevalence of any clinical manifestations upon comparing the co-infected patients versus those with just HAdv detected. The same data were found in other similar studies from Chile and Peru that found no difference in clinical severity between children co-infected with RSV-HAdv compared with children with HAdv monoinfection [Ampuero JS et al., 2012; Palomino MA et al. 2004; ].
While the identification methods (PCR, immunofluorescence, electron microscopy) were described, providing details about the sensitivity and specificity of these techniques could enhance the understanding of the identified viruses.
All these methods (PCR, immunofluorescence, and electron microscopy) a widely used and have known characteristics. We think that adding a description about these methods will causes of information overload. The sensitivity of PCR is 90-100% — with such a probability, the analysis will identify the pathogen if it really is in the sample. Compared to polymerase chain reaction, other methods are less sensitive. For example, the sensitivity of enzyme immunoassay is 50-70%, and the sensitivity of cultural studies is 60-80%.
Immunofluorescence is a widely used example of immunostaining and is a specific example of immunohistochemistry that makes use of fluorophores to visualize the location of the antibodies. Photons and light are dispersed when passing through the biological tissue of sufficient thickness. Therefore, light cannot penetrate deep into the tissues, with the maximal depth penetration by far-red light to be less than 100 μm (Berke IM et al., 2016). Most of the incoming light is scattered between the lipophilic membrane and aqueous environment of the tissue because each component has a different refractive index (RI) in the intact tissue. The tissue contains a little luminous source, each sending light in every direction and allowing us to observe a thin layer of tissue.
Electron microscopy (EM) allows fast visualization of viruses in a wide range of clinical specimens. Diagnostic EM has two advantages over enzyme-linked immunosorbent assay and nucleic acid amplification tests. After a simple and fast negative staining, EM allows fast morphological identification and differential diagnosis of infectious agents contained in the specimen without the need for special considerations and/or reagents. Nevertheless, EM has the disadvantage of being unsuitable as a screening method. Diagnostic EM can be performed when the agent is present in concentrations of a least 105 particles per mL (Almeida 1983; Gelderblom et al. 1991; Biel et al. 2004).
Explaining the clinical significance of the observed genomic variations within strains and how these variations might impact disease severity or treatment would add depth to the findings.
We did not indicate the influence of genotypes on the disease severity, because of limited sample numbers with known genotypes, which does not allow us to draw reliable conclusions.
Exploring the implications of similar virus titers across different genotypes on disease severity or transmission rates would be informative.
In this case, we indicate the infectious titers of viruses for cell culture, which may not be equivalent to the viral load in the patients from whom these samples were obtained, so it is difficult to draw conclusions about the effect of the infectious titer on disease severity or transmission rates.
Discussion:
The discussion lacks a clear introductory statement summarizing the objectives or the purpose of the study. The context of the study regarding the significance of understanding HAdV's epidemiology in acute respiratory infections requires expansion to provide a broader understanding.
We have added – «… In our study we analyzed certain molecular, epidemiological and virological characteristics of HAdV with C1, C2, C5, C89 and C108 detected among 3,190 samples from children 0-17 years old …”
The structure is disjointed, lacking clear subsections to segregate different aspects like epidemiological observations, virological characteristics, genomic analyses, and their implications. –
We have tried to provide the following explanation - `… However, such data do not allow us to make conclusions concerning the genotype prevalence, as we have not yet determined the complete genotype for all PCR-positive samples. Such work in the future will allow us to determine in more detail the epidemiology of adenoviruses and to assess whether there is a difference in clinical manifestations and disease severity. This will be of great importance for practical medicine…`
Points regarding the incidence among different age groups and the prevalence of various viruses could be presented more systematically for better comprehension.
We had previously published an article about the prevalence of various viruses and we did not repeat the same information in this one, but made the reference on it (Kurskaya, O.G.; Prokopyeva, E.A.; Sobolev I.A.; Solomatina M.V.; Saroyan T.A.; Dubovitskiy N.A.; Derko A.A.; Nokhova A.R.; Anoshina A.; Leonova N.; Simkina O.; Komissarova T.; Shestopalov A.M.; Sharshov K.A. Changes in the Etiology of Acute Respiratory Infections among Children in Novosibirsk, Russia, between 2019 and 2022: The Impact of the SARS-CoV-2 Virus. Viruses 2023, 15 (4), 934; https://doi.org/10.3390/v15040934).
The discussion frequently refers to earlier works (references [15, 16, 17]) without specifying the findings or context, making it hard to follow without referring to those sources directly.
We`ve tried to rewrite part of the text - `…In toto, as shown in our previous work, the assessment reports about etiological agents of ARVI during the period 2019-2022 that HAdV presented at a lower rate, following after HIFV, SARS-CoV-2, HRSV, and HMPV…`
Data, such as the incidence rates among different periods and age groups, lack precise statistical details or comparative analyses. The discussion lacks deeper analysis or interpretation of the observed trends. It presents data without adequately discussing the potential reasons behind variations in virus prevalence among different periods or age groups.
In this article we have mentioned – “…The lowest incidence of ARIs were registered among patients older than 7 (Table 1), as also determined in earlier works [19,20,21]. Those children 0-2 years old were the most sensitive to viral infections, with a 68% positive rate (1,248/1,836) (Chi-square test, P < 0.01). The most common etiological agents of ARI among children during 2019-2020 period were HIFV 28.7% (312/1088), during 2020-2021 period were HMPV 28% (316/1,130), and during 2021-2022 period were HRSV 20.3% (197/972). The most common etiological agents of ARI among children during the period 2019-2020 were HIFV 28.7% (312/1,088), during period 2020-2021 HMPV 28% (316/1,130), and during the period 2021-2022 HRSV 20.3% (197/972)…`
But the information deal with the rates among different periods and age groups already published (Kurskaya, O.G.; Prokopyeva, E.A.; Sobolev I.A.; Solomatina M.V.; Saroyan T.A.; Dubovitskiy N.A.; Derko A.A.; Nokhova A.R.; Anoshina A.; Leonova N.; Simkina O.; Komissarova T.; Shestopalov A.M.; Sharshov K.A. Changes in the Etiology of Acute Respiratory Infections among Children in Novosibirsk, Russia, between 2019 and 2022: The Impact of the SARS-CoV-2 Virus. Viruses 2023, 15 (4), 934; https://doi.org/10.3390/v15040934). And also there we discussed the potential reasons behind variations in virus prevalence.
Further analysis of the observed co-infection patterns or potential implications of co-infections on disease severity or clinical outcomes is absent.
Because we did not include clinical effects in the purpose of the study, we did not decide to discuss it in this Article. Besides, we have collected just a few samples with confirmed HAdV - only 127, that does not allow us to speculate on the topic about clinical implications.
The discussion lacks clarity regarding the significance of lower viral activity observed in comparison to other studies, requiring a more detailed explanation or possible implications of this difference.
As we have shown in vitro results, there are no differences in viral load among samples of HAdV that isolated in cell cultures (Table 3). Because we did not measure the amount of virus from patients directly, we don’t know were there any differences of viral activity.
The discussion of the genetic variations and phylogenetic analyses seems disconnected from the main narrative, making it hard to understand how these findings relate to the broader context of the study.
We`ve tried to rewrite the part Discussion part.
The findings regarding the unassigned strain (HAdVC/Novosibirsk/8.171V/2020) require a more detailed discussion regarding its implications and potential causes for the lack of classification.
By using the additional method that we have done for improving our results – the Simplot analysis - we have assigned to C108 genotype strain HAdVC/Novosibirsk/8.171V/2020, and therefore have rewritten the Text. Kindly, find new information in the Article (Fig. 6, Table 2 & 3, Fig. 7, Suppl. Table S1). Thus, no non-described recombinations in penton base, hexon and fiber ORFs among studied strains were found.
The conclusion is abrupt and lacks a summary of key findings. It should reiterate the most significant outcomes and their implications for the field of study.
The conclusion regarding the prevalence of genotypes lacks deeper insights or implications for future research or clinical practice.
The conclusion could be strengthened by suggesting future research directions, such as exploring the clinical significance of different genotypes, understanding the implications of co-infections, or further investigating unassigned strains.
The study's significant contributions, especially the identification of the HAdV-C89 genotype in Russia for the first time, could be emphasized more explicitly in the discussion and conclusion.
We have rewritten the conclusion completely - `… The study of the incidence of HAdV and other respiratory viruses, namely human influenza A and B viruses (HIFV), respiratory syncytial virus (HRSV), coronavirus (HCoV), parainfluenza virus (HPIV), metapneumovirus (HMPV), rhinovirus (HRV), bocavirus (HBoV), and SARS-CoV-2 among 3,190 hospitalized children showed that at least one of these respiratory viruses was detected in 74.4% of hospitalized cases, among which HAdV accounted for 4%, whereas co-infections with HAdV among 1.3 %. We obtained full genome sequences of twelve HAdVs, isolated in cell cultures. Genetic analysis has demonstrated that at least five? genotypes (C1, C2, C5, C89, and C108) of the HAdV virus were circulating among children (0-17 years old) with ARVI in Novosibirsk, Russia, during the period 2019-2022. We have revealed the circulation of 89 genotypes for the first time in Russia and suggest that frequent recombination among the different HAdV-C types might be an important driving force for the molecular evolution of HAdV-C. Further comprehensive and systematic monitoring, detection, and characterization of HAdV are necessary for future research and clinical practice. A future integrated approach which includes clinical data and obligatory full-genome analysis will provide the opportunity for the exploration of the clinical significance of different genotypes, understanding of the implications of co-infections, and the further investigation of yet unassigned strains. Data on differences in clinical manifestation and disease severity for certain of these genotypes will be of great importance for practical medicine…`
Extensive editing of English language required
We agree with reviewer’s assessment and tried to make proofreading.
Finally, we corrected the text and try to fix all the comments. Please see it again.

Round 2
Reviewer 1 Report
Comments and Suggestions for Authors
The revised manuscript is significantly improved. I still do not think that the EM studies are necessary but if the authors want to report them and the editorial office is OK with them, I am happy to accept them.
The following edits ARE required:
-On page 3, line 61, a "C" is missing before the number 89.
-On page 4, lines 87-91, the text is unclear and should be re-written. I suggest you start the sentence as follows: "the following RT-PCR kits were used:....."
-On page 4, line 92, the correct term is "CONSERVED" site.
-On page 8, line 177, the text needs to be edited to read "3.98% (127/3190) tested positive for HAdV"
-On page 10, line 207 the text should be edited to correct a major error: LOWEST (instead of highest) Ct values. The lower the Ct value, the higher the viral load.
On page 13, line 255, the text must be edited to read "...with extensive CPE and were processed for NGS with de novo assembly.
-On page 19, lines 337-339 is unclear. What does "in toto" mean?
The revised manuscript is significantly improved. I still do not think that the EM studies are necessary but if the authors want to report them and the editorial office is OK with them, I am happy to accept them.
The following edits ARE required:
-On page 3, line 61, a "C" is missing before the number 89.
-On page 4, lines 87-91, the text is unclear and should be re-written. I suggest you start the sentence as follows: "the following RT-PCR kits were used:....."
-On page 4, line 92, the correct term is "CONSERVED" site.
-On page 8, line 177, the text needs to be edited to read "3.98% (127/3190) tested positive for HAdV"
-On page 10, line 207 the text should be edited to correct a major error: LOWEST (instead of highest) Ct values. The lower the Ct value, the higher the viral load.
On page 13, line 255, the text must be edited to read "...with extensive CPE and were processed for NGS with de novo assembly.
-On page 19, lines 337-339 is unclear. What does "in toto" mean?
Author Response
Reviewer #1: Comments and Suggestions for Authors.
Dear Reviewer #1,
Thank you for your useful comments and questions that helped to improve the manuscript. Please see below each point for the detailed response on how we incorporated your feedback into the manuscript.
The revised manuscript is significantly improved. I still do not think that the EM studies are necessary but if the authors want to report them and the editorial office is OK with them, I am happy to accept them.
The following edits ARE required:
We`ve decided to demonstrated all information that we received in our study about the HAdV.
-On page 3, line 61, a "C" is missing before the number 89.
We agree with Reviewer and have added miss information.
-On page 4, lines 87-91, the text is unclear and should be re-written. I suggest you start the sentence as follows: "the following RT-PCR kits were used:....."
We agree with Reviewer and have re-written this phrase.
-On page 4, line 92, the correct term is "CONSERVED" site.
We agree with Reviewer and have corrected the word.
-On page 8, line 177, the text needs to be edited to read "3.98% (127/3190) tested positive for HAdV"
We agree with Reviewer and have re-written this phrase.
-On page 10, line 207 the text should be edited to correct a major error: LOWEST (instead of highest) Ct values. The lower the Ct value, the higher the viral load.
We agree with Reviewer and have re-written this phrase.
-On page 13, line 255, the text must be edited to read "...with extensive CPE and were processed for NGS with de novo assembly.
We agree with Reviewer and have re-written this phrase.
-On page 19, lines 337-339 is unclear. What does "in toto" mean?
Dear Reviewer, this formulation mean “in total…” and was proposed for our Manuscript by the translator. We have re-written it into “in total” now.
Finally, we corrected the text and try to fix all the comments. Please see it again.
Reviewer 3 Report
Comments and Suggestions for Authors
1. All figures resolution need to be improved
2. Phylogenetic trees need to be re-constructed/repeated.
Comments on the Quality of English LanguageModerate editing of English language required
Author Response
Reviewer #3: Comments and Suggestions for Authors.
Dear Reviewer #3,
Thank you for your useful comments and questions that helped to improve the manuscript. Please see below each point for the detailed response on how we incorporated your feedback into the manuscript.
- All figures resolution need to be improved
- Phylogenetic trees need to be re-constructed/repeated.
We have downloaded figures and phylogenetic trees in high resolution. Kindly find them in attached files. Also, we have asked translator agent to make proofreading of our Manuscript.
Finally, we corrected the text and try to fix all the comments. Please see it again.